# Sources, propagation and consequences of stochasticity in cellular growth

Philipp Thomas [1], Guillaume Terradot[2,3], Vincent Danos[4,5] & Andrea Y. Weiße [2,5,6]

Growth impacts a range of phenotypic responses. Identifying the sources of growth variation and their propagation across the cellular machinery can thus unravel mechanisms that underpin cell decisions. We present a stochastic cell model linking gene expression, metabolism and replication to predict growth dynamics in single bacterial cells. Alongside we provide a theory to analyse stochastic chemical reactions coupled with cell divisions, enabling efficient parameter estimation, sensitivity analysis and hypothesis testing. The cell model recovers population-averaged data on growth-dependence of bacterial physiology and how growth variations in single cells change across conditions. We identify processes responsible for this variation and reconstruct the propagation of initial fluctuations to growth and other processes. Finally, we study drug-nutrient interactions and find that antibiotics can both enhance and suppress growth heterogeneity. Our results provide a predictive framework to integrate heterogeneous data and draw testable predictions with implications for antibiotic tolerance, evolutionary and synthetic biology.

[1] Department of Mathematics, Imperial College London, London SW7 2AZ, UK. [2] SynthSys—Centre for Synthetic & Systems Biology, University of Edinburgh, Edinburgh EH9 3BD, UK. [3] School of Biological Sciences, University of Edinburgh, Edinburgh EH9 3FF, UK. [4] CNRS, École Normale Supérieure, Paris 75005, France. [5] School of Informatics, University of Edinburgh, Edinburgh EH8 9AB, UK. [6]Present address: National Institute for Health Research, Health Protection Research Unit in Healthcare Associated Infections and Antimicrobial Infections, Department of Medicine, Imperial College London, London W12 0NN, UK. Correspondence and requests for materials should be addressed to P.T. (email: p.thomas@imperial.ac.uk) or to A.Y.Wße. (email: andrea.weisse@imperial.ac.uk)

The rate at which cells accumulate mass and grow varies across isogenic cells[1–5]. Previous studies considered fluctuations in growth rate as one of the major drivers of phenotypic heterogeneity[4,6–8]. Yet the physiological origins of these fluctuations remain elusive so far. Growth laws characterise the typical behaviour of cell populations[9], for example, the scaling of average growth rate with cell mass in bacteria, or macromolecular composition[9–11]. These phenomenological relations can give important insights into the population-average behaviour, but may not translate to an understanding of individual cell responses[3].

There is substantial evidence that cellular noise sources are diverse and may propagate in a systemic way. A recent experimental study showed that fluctuations in the expression of enzymes caused considerable variation in the growth rate of single cells, which then fed back onto their expression and that of other genes[1]. Cell-to-cell differences stem from fluctuations intrinsic to biochemical reactions[12]. Some of these reactions, particularly those that drive cell growth, affect many other intracellular processes, and so cellular responses can vary even under constant conditions[13]. Gene expression, for example, is inherently stochastic at the single-cell level[12]. It is less clear though how such variation affects other intracellular processes[14,15], and how it translates to phenotypic differences and cell fitness.

Models can help identify potential sources of fluctuations and understand how they propagate to cause phenotypic variation. There are various approaches to model cellular growth. One is to invoke growth rate optimisation[16–18], another to consider the coordination of growth with gene expression[9], or to combine the two approaches[19]. Such approaches have been used to model static cell-to-cell variation by imposing parameter variability onto the model behaviour[20,21]. The sources of growth variations, however, remain unclear, and also how to adapt the models to explain cell responses that fluctuate over time.

We present a stochastic model of single-cell bacterial dynamics to predict the growth rate of individual cells. Our description of cells is based on biochemical kinetics, which accounts for stochastic fluctuations in cellular mechanisms giving rise to heterogeneous responses. In this context, the magnitude of fluctuations results from the abundance of key molecular players[22], and so we can predict emergent growth variations rather than impose them onto the model behaviour.

The model builds upon recent insights into population-average growth via a mechanistic description that explains Monod growth and empirical relations between growth rate and ribosomal contents from the interplay of nutrient uptake, metabolism and gene expression[9,23]. These processes are constrained by cellular trade-offs such as a finite transcriptome and proteome, as well as a limited pool of ribosomes and cellular resources. Here we consider the finite number of intracellular molecules produced over a cell cycle and so explicitly account for biomass production and its corresponding stochastic dynamics. We further integrate this approach with a model of bacterial cell-cycle control[24,25], supported by recent experiments[4,26], to quantitatively predict emergent growth and division dynamics in single *E. coli* cells.

Along with the cell model we present a theoretical framework to approximate stochastic growth and division dynamics. The framework is applicable to models of reaction–division systems at large. It enables closed-form computation of model statistics, such as mean and variance of variables over time, and thus allows efficient parameter estimation from single-cell data alongside a systematic decomposition of the sources of growth variation and model exploration via parameter sensitivity.

Our modelling approach, in combination with the developed approximation, allows us to statistically characterise the macromolecular composition, growth rate and mass of single cells. It recovers several empirical responses at the population- and single-cell level, thus providing substantial validation. We quantify the contributions of different noise sources to observed growth rate fluctuations and analyse their propagation. We identify dynamics of mRNAs coding for nutrient transporters and enzymes as a major source of growth rate fluctuations. Fluctuations in growth rate in turn transmit noise to other processes[1], for example, via ribosomes, as has been hypothesised previously[27,28]. Our analysis of cell responses to translation-inhibiting antibiotics further indicates a strikingly complex dependence of growth heterogeneity on environmental conditions, which may pinpoint strategies to avoid drug tolerance.

## Results

**A stochastic model of single-cell growth.** Models that coordinate growth and division in single cells need to integrate many processes at different scales. We take a hybrid approach to model growth and division by combining deterministic DNA replication with stochastic biomass production (Fig. 1a). As cellular protein content dominates biomass[29], we assume that total translation rate sets the rate of biomass production[9,23]. In a single cell, translation is coupled to processes that fuel and drive gene expression. Since these processes are stochastic, growth rate varies over time and from cell to cell. We use a bottom-up approach that describes the dynamics of a coarse-grained cellular composition based on stochastic biochemical reactions, which comprise transcription, translation, ribosome binding, mRNA degradation and metabolism (see Methods).

The model describes the accumulation of proteomic mass—split into sectors containing transporters ($t$), catabolic enzymes ($e$), ribosomes ($r$) and housekeeping proteins ($q$)—along with the corresponding transcriptome, ribosome-mRNA complexes and a resource species. The resource is a coarse-grained variable describing the collection of molecules that fuel biosynthesis, for example, energetic molecules such as ATP and NAD(P)H or charged tRNAs, depending on the nutrient limitations under consideration. For simplicity, we focus on one transporter and one enzyme species representing metabolic bottlenecks.

At the single-cell level, we account for cell divisions using the Cooper–Helmstetter model, in which cells divide after a constant period following initiation of DNA replication[24]. Replication cycles couple to growth through initiation at a fixed concentration of replication origins[25], allowing for parallel replication rounds. As a consequence, the cell mass at initiation depends on the number of ongoing replication rounds in a given growth condition (Methods).

To capture the stochastic dynamics of the model we focus on a lineage description that tracks a single cell over various replication and division cycles. At division, intracellular molecules are partitioned randomly between the two daughter cells, and we retain information about only one daughter cell[30]. We account for asymmetric cell division, as for instance due to inaccurate positioning of the division septum[2,3,31], and assume that molecules are partitioned according to the inherited volume fraction of the daughter cell (Supplementary Note 1).

Stochastic simulations illustrate the dynamic propagation of fluctuations (Fig. 1b): the stochastic synthesis, degradation and partitioning of mRNA molecules lead to slow fluctuations in the

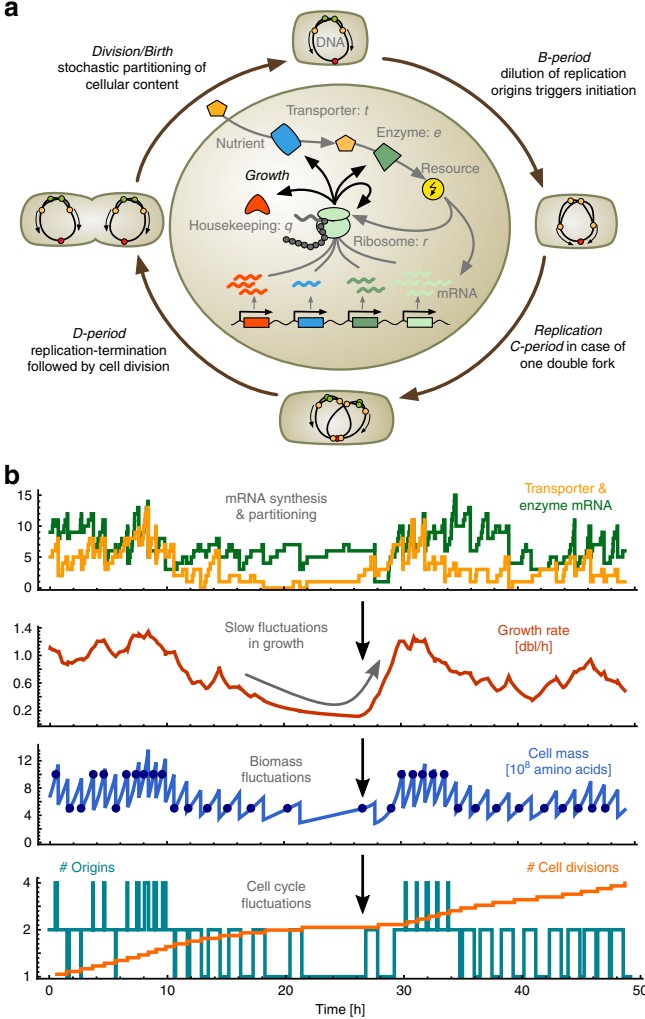

**Fig. 1** Stochastic model of single-cell growth. **a** The outer cycle illustrates the cell cycle model based on the Cooper–Helmstetter model of bacterial replication. We assume initiation of a new round of replication at a fixed concentration of DNA-origins, analogous to a fixed initiation mass per DNA-origin[25], thus growth dynamics schedule the replication events and are determined by the intracellular model (inner circle). The latter describes import and metabolism of resources, and how they fuel gene expression, where the rate of protein-biosynthesis determines growth. Stochasticity of cellular dynamics is a result of the intrinsic stochasticity of the various reactions and the random partitioning of the cellular content at division. **b** Stochastic simulations illustrate the propagation of intrinsic fluctuations in single cells: mRNAs are synthesised at low numbers per cell (yellow & green lines), which affects protein production and so growth rate (red line). Fluctuations in growth lead to temporal variations in cell mass that can span several cell cycles (blue line), causing fluctuations in the number of replication origins (teal line), in the mass at initiation (filled circles), and consequently in cell divisions (orange line)

growth rate and in turn to variations in biomass production and division times. Simulating the stochastic model is computationally expensive due to the large number of molecules produced per division cycle. We therefore developed an approximation method that gives quantitative insights into a whole class of cell models involving coupled reactions and divisions.

**Stochastic analysis of reaction–division systems.** Consider a generic reaction–division system composed of $N$ intracellular species with molecule numbers $\mathbf{x} = (x_1, .., x_N)^T$. The macromolecular composition of a single cell determines its mass via

$$M = \mathbf{m}^T \mathbf{x}. \qquad (1)$$

The components of the vector $\mathbf{m}$ denote the mass of individual molecules. At constant density this measure is directly related to cell size. The corresponding intracellular concentrations

are given by

$$\mathbf{X} = \frac{\mathbf{x}}{M}. \qquad (2)$$

For an intracellular reaction network comprising $R$ reactions with stoichiometric matrix $\nu$, the cell growth rate

$$\lambda(\mathbf{X}) = \mathbf{m}^T \nu \mathbf{f}(\mathbf{X}), \qquad (3)$$

can be obtained analytically from the vector of reaction rate functions $\mathbf{f}(\mathbf{X})$ (see Supplementary Note 2). Because intracellular reactions are stochastic, the concentrations $\mathbf{X}$ fluctuate over time and hence the growth rate is a stochastic process.

We describe the dynamics of intracellular concentrations, cell mass and its growth rate using a continuous approximation of biochemical reactions and dilution from biomass production, combined with discontinuous cell divisions. The set of coupled

Langevin equations for the jump-diffusion process is

$$
\begin{aligned}
\mathrm{d}\mathbf{X} =\ & \underbrace{\nu \mathbf{f}(\mathbf{X})\mathrm{d}t}_{\text{biochemical reactions}} + \underbrace{\frac{1}{M^{1/2}}\sum_{r=1}^{R}\nu_r\sqrt{f_r(\mathbf{X})}\,\mathrm{d}W_r(t)}_{\text{noise from biochemical reactions}} \\
& -\underbrace{\lambda(\mathbf{X})\mathbf{X}\mathrm{d}t}_{\text{dilution}} - \underbrace{\frac{\mathbf{X}}{M^{1/2}}\sum_{r=1}^{R}\mathbf{m}^T\nu_r\sqrt{f_r(\mathbf{X})}\,\mathrm{d}W_r(t)}_{\text{noise from biomass synthesis}} \quad (4) \\
& +\underbrace{\frac{1}{M^{1/2}}\boldsymbol{\xi}_{D(t)}\mathrm{d}D(t)}_{\text{partitioning noise}},
\end{aligned}
$$

$$
\begin{aligned}
\mathrm{d}\ln M =\ & \underbrace{\lambda(\mathbf{X})\mathrm{d}t}_{\text{growth}} + \underbrace{\frac{1}{M^{1/2}}\sum_{r=1}^{R}\mathbf{m}^T\nu_r\sqrt{f_r(\mathbf{X})}\,\mathrm{d}W_r(t)}_{\text{noise from biomass synthesis}} \\
& -\underbrace{\frac{1}{2}\left(1+\eta_{D(t)}\right)\mathrm{d}D(t)}_{\text{cell divisions}} + \underbrace{\frac{1}{M^{1/2}}\zeta_{D(t)}\mathrm{d}D(t)}_{\text{partitioning noise}}. \quad (5)
\end{aligned}
$$

The process $D(t)$ counts the number of divisions (see Fig. 1b, Supplementary Note 1), independent Gaussian white noises $W_r(t)$ describe the intrinsic variability of the intracellular reactions, the random variables $\xi_D$ and $\zeta_D$ introduce noise from partitioning of molecules at division, and $\eta_D$ accounts for variation in the inherited volume fraction. Specifically, we consider binomial partitioning of all species that satisfies $E\left[\xi_{D,i}\xi_{D,j}|\mathbf{X}\right] = X_i\left(\delta_{ij} - \frac{m_i m_j X_j}{\sum_l m_l^2 X_l}\right)$ and $E\left[\zeta_D^2|\mathbf{X},M\right] = \frac{1}{4}\sum_{l=1}^{N}m_l^2 X_l$ (see Supplementary Note 2 for a detailed derivation). The vector $\nu_r$ denotes the $r$th column of $\nu$. The concentration process, Eq. (4), satisfies the conservation $\mathbf{m}^T\mathbf{X} = 1$ and is coupled to the mass process, Eq. (5), except in the deterministic limit (when $M$ is large).

Between cell divisions ($\mathrm{d}D = 0$), Eq. (5) yields the instantaneous growth rate, which has two contributions: $\lambda(\mathbf{X})$, a deterministic function of the fluctuating concentrations, and a second random term from the mass-producing reactions. For biologically relevant situations, the second term is negligible due to averaging over a large number of such reactions occurring between divisions (Supplementary Note 2). In the absence of growth, that is, when all reactions are mass-conserving ($\mathbf{m}^T\nu_r = 0$), Eq. (4) reduces to the standard chemical Langevin equation[32].

To gain further analytical insight we developed a small noise approximation[33,34] of Eqs. (4) and (5). It allows us to compute mean concentrations and growth rate by solving a coupled system of ODEs in steady-state conditions. Concentration fluctuations lead to growth rate variations that can be computed in closed form (Methods). The method provides accurate estimates of the first two statistical moments (Fig. 2) enabling efficient inference of model parameters, which is typically infeasible using stochastic simulations[35]. We used it to infer parameters of our stochastic cell model, fitting predictions to experimental bulk measurements of ribosomal mass fractions[9] and single-cell fluctuations in growth rate[1,3], both measured across a range of growth conditions (Fig. 2, Methods, Supplementary Note 3). We discuss the results and predictions drawn from this inference in the following.

**Condition-dependence of growth in single cells**. The macro-molecular composition of E. coli is growth-rate dependent, and we ask whether the cell model is consistent with several bacterial

growth laws describing these relations. Our model predicts that mean cell mass increases exponentially with mean growth rate, the Schaechter–Maaløe–Kjeldgaard growth law[10], as a consequence of the coupling of DNA replication to growth[25]. Moreover, unit size[36], in terms of mass per number of origins, is invariant across growth conditions (Methods, Eq. (12)). If we compare the theoretical unit size with recent measurements of unit volumes in E. coli[36] (Fig. 2a, first panel), the model predicts a protein density of $12 \times 10^8$ aa/μm$^3$, well in line with literature values[37]. With this density estimate, model predictions closely match cell size measurements of two different data sets[2,36] (Fig. 2a, second panel). The inferred model further recovers ribosome abundances over the experimental range of growth rates[9] (third panel) and predicts transcriptome and proteome sizes that are in qualitative agreement with experimentally observed values[38,39] (fourth panel, and Supplementary Fig. 3b).

Recent experiments suggest a universal behaviour of cell size and doubling time distributions when rescaled by their mean[2,3], indicating that growth conditions primarily affect the mean cell size, doubling time, and growth rate. Our model reproduces this dependence for the cell size and added mass in intermediate to fast growth conditions (Fig. 2c). For conditions slower than those measured in ref. [3] we observe a breakdown of this scaling. In those conditions, our model predicts an increase in cell size variability with growth rate (Fig. 2b) due to a shift from single to parallel rounds of replication[4,40] (Supplementary Fig. 3a). Such increases in cell size variation have indeed been observed in cells grown in the mother machine[2,40]. We observe no scale invariance for doubling times and growth rates (Fig. 2d), indicating that their cell-to-cell variations are condition-dependent rather than universal.

Recent data indeed suggest a condition-dependence of growth rate fluctuations in single E. coli cells[1,3] (Fig. 2e). In line with these observations, our model predicts growth variations to decrease with mean growth rate. This dependence is well captured by the developed approximations and stochastic simulations (Methods). Our model predicts that growth rate saturates for increasing nutrient qualities[9,23], consistent with Monod growth, and that fluctuations diminish as the mean growth rate approaches its maximum. We confirmed that the behaviour is robust to parameter variations (Supplementary Fig. 4). This is not because intracellular reactions stop fluctuating, but because growth saturates and fluctuations in resource levels no longer translate to growth variability. Ribosomes moreover exhibit little fluctuations as they are highly abundant (cf. Eq. (9) and Supplementary Fig. 5).

**Sources of growth rate fluctuations**. Our model allows us to investigate the sources of phenotypic variations. We developed a noise decomposition (Supplementary Note 2 and ref. [41]) to reveal the contribution of each reaction and partitioning at division to the overall noise level. We find that transcription and the partitioning of cellular contents at division are the major determinants of growth heterogeneity that together explain most variation across all growth conditions (Fig. 2e). Degradation of mRNA becomes important only at slow growth (<1 dbl/h), because for faster growth most mRNAs are bound to ribosomes and shielded from degradation. Nutrient uptake and metabolism, in turn, yield negligible contributions because nutrients are highly abundant (Supplementary Fig. 5). Similarly, effects of noise in translation are mostly negligible, due to the large number of proteins synthesised during a cell cycle, and only contribute to growth variations at very small growth rates (<0.1 dbl/hr). In such conditions, however, regulatory mechanisms as involved in starvation are expected to take effect which our model does not describe.

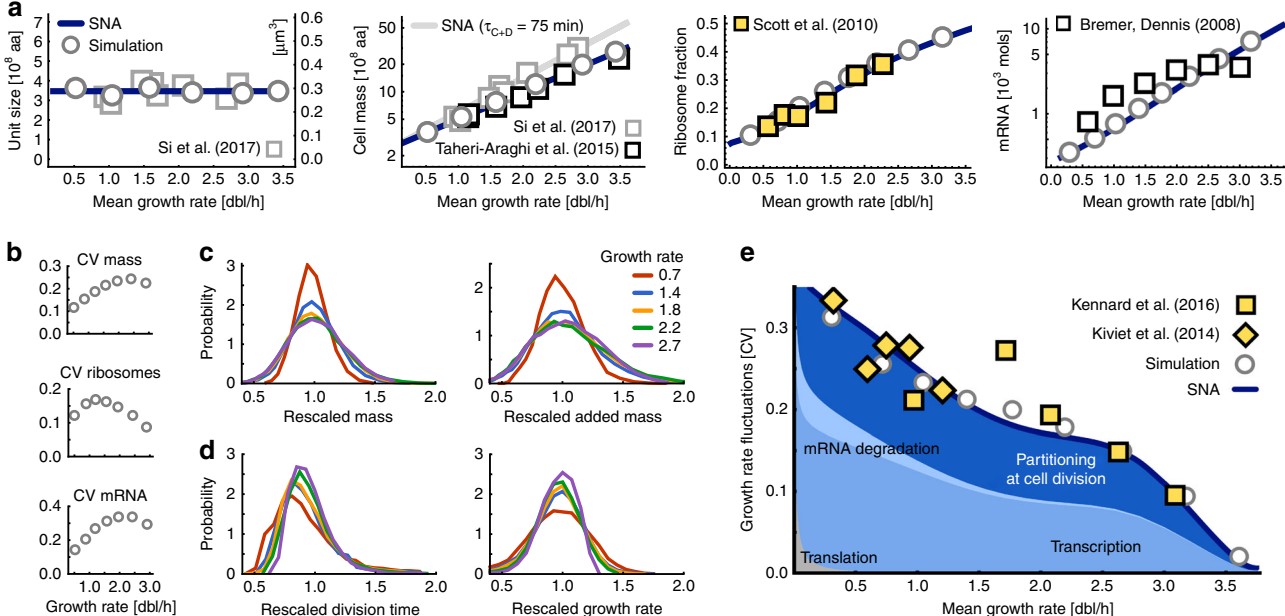

**Fig. 2** Stochastic model predicts condition-dependence of growth in single cells. **a** Our model recovers bacterial growth laws. Cell mass per number of origins (unit size) is constant in all growth conditions. Absolute cell mass, ribosome content and total mRNA numbers per cell increase with average growth rates. Measurements used for model parameterisation (yellow markers), measurements validating model predictions (white markers); stochastic simulations (circles) validate the approximations (SNA, lines). A longer $C+D$-period of 75 min[36] yields higher mass (grey line). Consistent with Scott et al.[9] and Kennard et al.[3], we changed growth rate by varying nutrient quality ($n_s$ in Eq. (8)). Varying transporter or enzyme levels as in Kiviet et al.[1] has a qualitatively similar effect (Supplementary Fig. 4). **b** Fluctuations in cell mass, measured by the coefficient of variation (CV), initially increase as a function of average growth rate. Fluctuations of ribosomal mass fraction are of the order of 10–20%, and those of total mRNA concentrations largely follow the trend of the mass CV. **c** Single-cell distributions of cell mass at birth and mass added between birth and division are invariant when rescaled by their means. For intermediate to fast growth conditions (1.4–2.7 doublings per hour) distributions collapse nearly perfectly, consistent with the stable CV in this growth regime (**b**). Slowly growing cells (0.7 red line) deviate from this universal behaviour. **d** Rescaled distributions of doubling times and growth rates broaden gradually with decreasing medium quality, i.e. the quantities are condition-dependent at the single-cell level. **e** Our model quantitatively recovers variations over the whole range of experimentally accessible growth rates. Stochastic simulations (grey circles) and the small noise approximation (SNA, solid blue line) predict that fast growing cells display less growth variability than slow-growing cells, consistent with experimental observations (diamonds[1], squares[3]). Colours indicate the contributions of different cellular processes to growth variations: synthesis, degradation and random partitioning of mRNAs at cell division. Contributions from other processes such as protein translation are overall small (grey area)

The dominant contribution to growth variation stems from the synthesis and removal of transporter mRNAs (Fig. 3a). This suggests that nutrient uptake limits growth rate, consistent with estimated catabolic rates exceeding those of nutrient uptake (Supplementary Table 1). Since catabolic rates can be tuned by cofactors[42], we wondered whether limiting catabolic turnover could affect growth fluctuations. When catabolic turnover is slower than nutrient uptake, indeed, growth variations are due to the transcription and removal of enzyme mRNAs rather than transporter mRNAs (Fig. 3b). The total size of growth fluctuations remains largely unaffected by whether uptake or catabolism limits growth. Surprisingly though, when both nutrient uptake and catabolic turnover are simultaneously rate limiting, both mRNA species contribute fluctuations but growth variability drops, suggesting a noise cancellation effect (Supplementary Fig. 6b). Operon organisation of the corresponding genes does not affect our predictions, except that the simultaneous limitation by transport and catabolism does not lead to noise cancellation (Supplementary Fig. 6).

We asked whether the observed noise levels may also be explained by fluctuations in the expression of ribosomes rather than metabolic components. We therefore considered an alternative model with continuous supply of resources, which allowed us to study a potential ribosome limitation on cell growth in the absence of other limitations (Supplementary Note 4). This reduced model is indeed able to explain both the condition-

dependence of mean ribosome concentrations and growth rate fluctuations based on fluctuations in lowly abundant r-mRNAs, and consequently, an overall smaller pool of mRNAs (Fig. 3c, Supplementary Fig. 7b, c). This implies that non-rate-limiting q-mRNAs are also lowly abundant, and so contribute to growth fluctuations. To discriminate which of the limitations we considered provides a biologically plausible explanation of the data, we examined absolute mRNA abundances as they present the sources of variations[13]. In fact, r-mRNAs belong to the most abundant mRNAs in the cell ranging from $10^2$ to $10^3$ molecules on average, depending on growth conditions[38,43] (Supplementary Fig. 7e), much higher than the numbers predicted by the reduced model with ribosome limitations (Supplementary Fig. 7f).

In contrast, our full cell model, which considers nutrient and metabolic limitations, predicts molecule numbers that are in quantitative agreement with measured r-mRNA abundances (Supplementary Fig. 7f). Further, predicted mean abundances of transporter and enzyme mRNAs vary in different conditions between 3 and 9 copies per cell, with ~9000 molecules of the corresponding proteins (Supplementary Fig. 5). Compared to that, natural abundances are between $10^{-3}$ and 1 mRNAs per gene while proteins are more abundant with 1 to $10^3$ molecules, with products of essential genes occurring at higher numbers[44]. This suggests that transporter and enzyme species in our model are consistent with lumped groups of enzymes rather than a single rate-limiting species. In summary, the data suggest that,

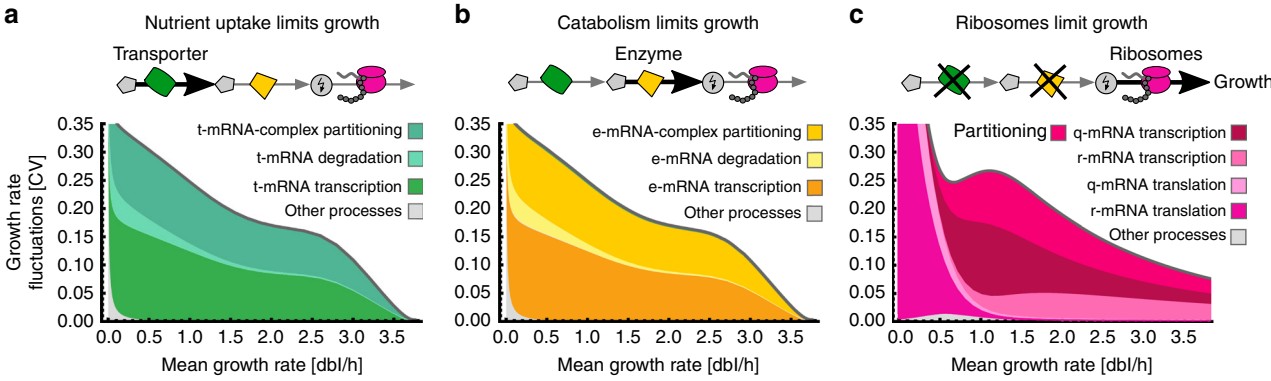

**Fig. 3** Growth limitations determine the sources of growth fluctuations. **a** Fluctuations in transcription of transporter mRNAs and their stochastic partitioning dominate growth variability when nutrient uptake is growth-limiting (import rate < enzymatic rate, see Supplementary Table 1). Other processes such as translation of proteins are largely negligible. **b** When catabolism limits growth (import rate > enzymatic rate), fluctuations in enzymatic mRNAs instead explain most of the observed variation. We vary catabolic turnover rates $v_m$ relative to nutrient uptake rate $v_t$. **c** In a reduced model with constant supply of resources, ribosome production limits growth. Growth variability results from transcription, translation and partitioning of lowly abundant r- and q-mRNAs

rather than ribosomes, metabolism and nutrient-uptake limit growth. We assumed the latter for all analyses that follow.

Our model does not include specific regulatory interactions. We therefore explore potential effects of regulation by varying parameter values in response to different growth conditions. We analysed the sensitivity of mean growth rate and its fluctuations to all model parameters in different growth conditions (Supplementary Fig. 4).

Interestingly, DNA replication has a profound impact on growth fluctuations but not on the mean growth rate. A higher concentration of origins at replication amplifies heterogeneity because cells become smaller and express lower absolute levels of mRNA. Similarly, delays in replication and division (C and D-periods in Fig. 1a), which imply larger cells, attenuate growth fluctuations in moderate to fast conditions but to a lesser extent in slow conditions (<1 dbl/hr), where it is known to vary[4,36].

Growth heterogeneity also exhibits high sensitivity to transcription rates. Inhibition of ribosome production, as for example via ppGpp, attenuates fluctuations, however, at the cost of reduced growth in fast conditions (>2.5 dbl/hr). Because we consider uptake limitation, fluctuations also decrease when upregulating transporter expression, for instance via cAMP-CRP regulation, exerting a similar effect to changing the efficiency of nutrient uptake or nutrient quality. Strikingly, stronger autoregulatory feedback on q-protein expression increases mean growth but dampens fluctuations, especially in slow growth conditions (<1 dbl/hr), because it relieves resources such that limiting components can be expressed at higher levels. Many of E. coli's proteins regulate their own expression[45], and our results suggest that the ubiquity of negative autoregulation[46] may be advantageous to reduce growth heterogeneity.

**Propagation of fluctuations**. We further ask how stochastic fluctuations propagate to growth, and how this affects the macromolecular composition of a single cell. Since growth rate feeds back onto all intracellular concentrations via dilution, it is not straightforward to determine the flow of information. We use cross-correlation between growth rate and different intracellular concentrations, computed from stochastic simulations, to quantify the propagation of fluctuations. These correlations measure the similarity of a small number of lumped macrovariables—such as total ribosome concentrations or growth rate—at different instances of time and thus reveal their temporal order. They do

not necessarily imply a causal interaction between molecules, which is set by the underlying biochemical reactions. We quantify noise propagation by the lag, which is the time of maximal correlation and measures the delay between variables. Upstream components have positive lags with growth rate suggesting that they transmit fluctuations to growth, and growth can either increase or dilute downstream components. Since the sources of growth fluctuations promote growth, upstream components should correlate positively with growth but correlations of growth with downstream components may be positive or negative.

We observe a strong positive correlation of transporter mRNA concentrations with growth rate at later times (Fig. 4a), consistent with our previous finding that their fluctuations are the major source of growth variations. Ribosome concentrations also correlate positively with growth, consistent with the increase of mean levels with growth rate (compare Fig. 2a). Interestingly though, they correlate at a negative delay, suggesting that fluctuations in growth propagate to ribosomes but ribosome fluctuations do not contribute substantially to growth variability. Other species such as enzyme mRNAs correlate negatively at a negative delay, indicating their concentrations are mainly affected by dilution, a relation that we observe more generally for their corresponding enzymes and also for q-mRNA across all conditions.

To estimate the propagation of fluctuations in the upstream and downstream processes of growth we consider the delay between any pair of groups (Fig. 4b). The intuition behind this is that a minimal positive delay suggests the species that first senses a fluctuation, which it then passes on to the next species. We illustrate this flow of information in a directed graph, where edges indicate the minimal delay relation between groups of species and edge weights their correlation (Fig. 4c). Note that the minimal delay graphs only reflect the dominant paths of noise propagation and do not exclude potential weaker correlations with other components.

Consistent with growth limitations, the corresponding fluctuations in either transporter or enzyme mRNAs are the source of growth rate variation that propagate via their respective protein levels and resources to growth. When transporters and enzymes are co-expressed from an operon, as in ref. [1], their common mRNA is the dominant source of growth variations such that the noise propagation cannot be distinguished between different limitations. Other components are downstream of growth rate, steadily across different growth conditions (Supplementary

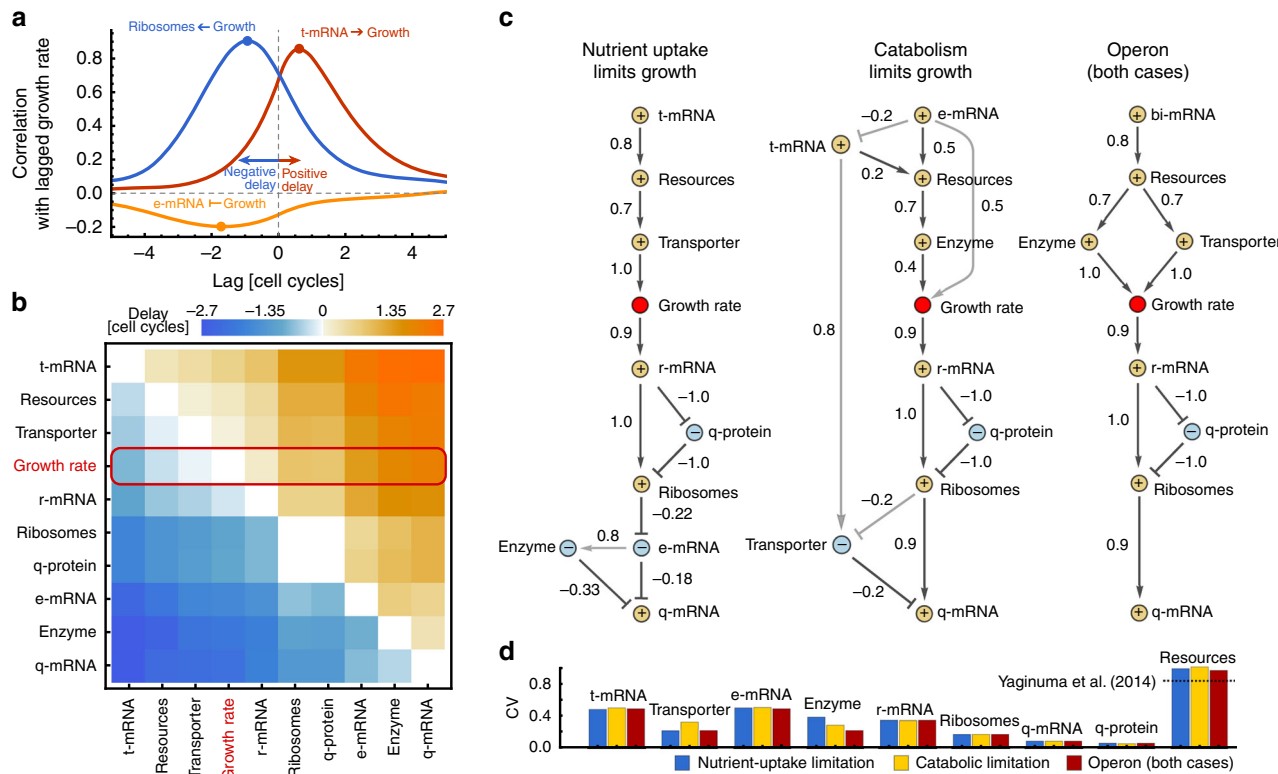

**Fig. 4** Cross-correlation analysis reveals the propagation of fluctuations. **a** Concentrations of transporter mRNA (t-mRNA) correlate positively with growth rate at later times (red line, maximum correlation at positive lag) indicating that they increase growth rate. Ribosome concentrations correlate positively with growth at earlier times (blue line, maximum correlation at negative lag) indicating that they are increased by growth rate. Concentrations of enzymatic mRNA (e-mRNA) correlate negatively, indicating that they are mainly diluted by growth (yellow line, minimum correlation at negative lag). **b** Pairwise delays between cellular components and growth rate (red box), ordered by their delay with respect to growth rate, computed in moderate growth conditions (1.4 dbl/hr). **c** Minimal delay graphs illustrate the information flow under growth limitations with comparable growth rates. A dark arrow from species A to B indicates that B has minimal delay from A, suggesting that species B is the first to receive fluctuations from species A. Arrows denote positive correlations at the delay, T-arrows negative correlations, and labels denote the delayed-correlation coefficient. Light arrows indicate components with second smallest delay whenever these are not reached through subsequent steps. The graph reveals cellular components up- and downstream of growth rate, i.e. those that affect growth and those affected by growth. When nutrient-uptake is limiting growth, t-mRNA act as a source of fluctuations, while for catabolic limitation e-mRNA are the dominant source (cf. Fig. 3). The corresponding proteins are upstream of growth and transmit fluctuations to growth rate. When transporters and enzymes are co-expressed from an operon the different limitations are indistinguishable. In all cases, q-mRNAs act as a sink due to their negative autoregulation, q-proteins are mainly diluted (nodes labelled with − correlate negatively with growth) while most species increase with growth (nodes labelled with + correlate positively with growth). **d** Fluctuations (CV) in concentrations of the transcriptome, proteome and resources are comparable across the considered cases. The dashed line indicates measured fluctuations in intracellular ATP[47]

Fig. 8a), suggesting that they are affected by growth. Only at high growth rates ribosomal transcripts—but not their proteins—are upstream of growth, because in these conditions fluctuations in ribosomes rather than in resources dominate noise in growth rate (Supplementary Fig. 8, cf. Eq. (9)). Interestingly, q-mRNAs act as noise sinks across conditions as they are subject to negative autoregulatory control.

Highly abundant species have consistently lower noise levels (Fig. 4d, Supplementary Fig. 5) except resources, which display an extremely high variability, likely due to their central role in many cellular reactions. This prediction is quantitatively confirmed by recent experiments showing that ATP levels in *E. coli* vary up to 80%[47] (compare Fig. 4d). The analysis shows that growth rate affects a large number of downstream components, which may include, for example, transcription factors controlling stress responses or other phenotypic switches. Our results therefore underline the central role of growth rate as a source of phenotypic heterogeneity[1].

**Growth heterogeneity in response to antibiotics**. We next examine bacterial responses to antibiotic treatment. The common

route to assess the efficacy of drugs is by establishing the dose-dependence of growth rate[48]. Growth heterogeneity, however, gives rise to antibiotic tolerance, which allows individual cells to survive treatment through non-genetic mechanisms[49–51]. Surviving cells can then develop and pass on mutations that confer resistance, and so growth heterogeneity contributes to the rise of antibiotic resistance.

Assuming that chloramphenicol imposes limitations on ribosome availability by inactivating ribosome complexes (Methods), our model correctly predicts average drug responses without re-fitting (Fig. 5a). We therefore used the model to quantitatively map both average growth rate and growth heterogeneity to combinations of nutrient and antibiotic regimes.

Unsurprisingly, the model predicts that average growth rate increases with nutrient quality and decreases with antibiotic dose (Fig. 5b). Growth heterogeneity, however, exhibits a complex landscape in response to combined nutrient and ribosome limitations (Fig. 5c), in contrast to the response under the individual limitations (Fig. 3a, c). For all nutrient conditions growth heterogeneity rises steeply at high drug concentrations. But only in very rich conditions, where growth rate saturates,

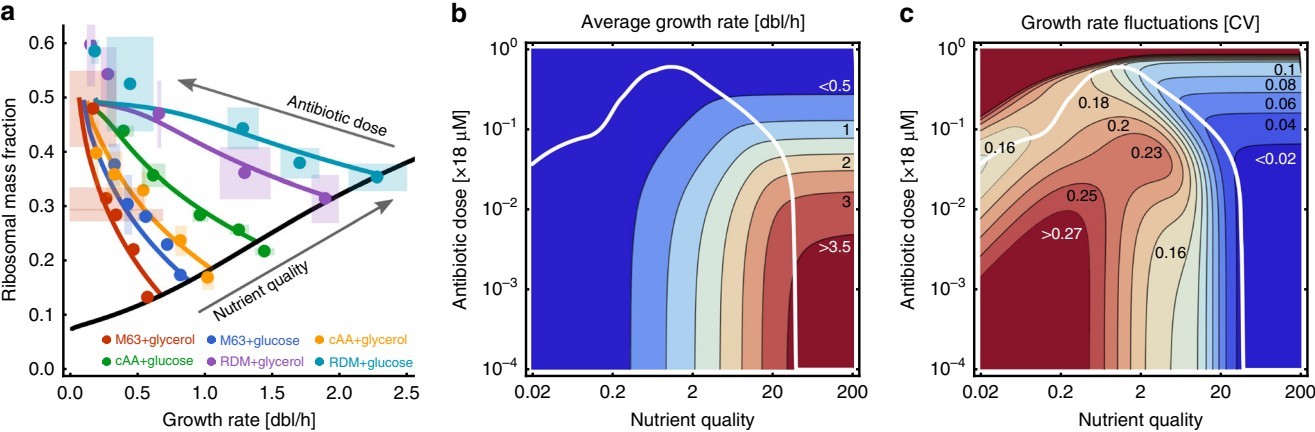

**Fig. 5** Condition-dependence of antibiotic responses. **a** Ribosomal content per cell as a function of average growth rate after treatment with chloramphenicol. Predictions (solid lines) are in quantitative agreement with experimental data[9] (dots, shaded areas denote standard deviations over replicates and colours denote different nutrient conditions). **b** For any given nutrient condition the average growth rate is predicted to decrease monotonically with antibiotic dose. **c** Growth heterogeneity is predicted to be highly complex in a both nutrient- and dose-dependent manner. In nutrient-rich conditions growth heterogeneity increases with antibiotic dose, while in intermediate and poor conditions the response is non-monotonic. Over a large range of nutrient conditions there exists a non-zero drug dose that minimises growth heterogeneity (solid white line, **b**, **c**)

growth heterogeneity increases monotonically with antibiotic dose, consistent with the inverse relation of average growth and heterogeneity (cf. Fig. 2d). In all other nutrient conditions, growth heterogeneity is a non-monotonic function of dosage: In medium-to-rich nutrient conditions, heterogeneity first peaks and then dips before the final rise. In low-to-medium nutrient conditions, the final rise is preceded by a drop in growth heterogeneity at intermediate doses.

Our predictions suggest that avoiding regimes of high growth heterogeneity may be achieved in different ways depending on the location of an infection. For example, it may be possible to treat infections in low-to-medium nutrient conditions, such as the urinary tract or blood, with a dose that minimises heterogeneity (Fig. 5c, white line). This would require more care for infections of richer nutrient environments, such as the gut, where regimes of increased heterogeneity should be avoided. The predictions further suggest that infections of very rich environments cannot be treated with an overall reduction of growth heterogeneity. Notably though, heterogeneity is mostly low in these conditions, and so an ideal dose may be chosen high enough to sufficiently inhibit growth but low enough to avoid regimes of significant heterogeneity. Alternatively, treatment efficacy may be manipulated by changing the environment of the pathogen, for example, by constraining diet.

## Discussion
We presented a stochastic cell model to predict growth and division dynamics in single bacterial cells. Our model makes detailed predictions of growth rate, size and macromolecular composition of cells in response to complex environments such as nutrient conditions and drug doses. In contrast to previous approaches, we also predict how phenotypic heterogeneity arises from molecular mechanisms in single cells where physiological responses emerge from intrinsic fluctuations of biochemical reactions.

We quantitatively recover levels of growth heterogeneity that have been measured in individual bacterial cells, and predictions are in good agreement with absolute transcriptome and proteome levels (Fig. 2a and Supplementary Figs 3 and 7f) per cell as reported in bulk measurements. We observe scale invariance of several macroscopic quantities over the range of experimentally reported conditions, indicating that cell-to-cell variations are

independent of these growth media. Our results moreover suggest that this scale invariance breaks down if tested over a broader range of growth conditions. In particular, we predict an increase of cell size variability with growth rate, in agreement with recent experiments[2,40].

We presented a general framework to analyse stochastic reaction–division dynamics. Our theoretical analysis enabled us to dissect the contributions of different biochemical processes to the observed growth heterogeneity. Specifically, we identified fluctuations in the synthesis, partitioning and degradation of mRNAs coding for proteins involved in metabolism as the major source of growth heterogeneity. The prediction is in line with observations that mRNAs of essential genes can naturally be present at low molecule numbers per cell[44] and that fluctuations in enzyme expression can cause growth rate variation[1]. In fact, expression of glucose transporters in *E. coli* has been reported to be highly heterogeneous[52].

In agreement with experiments, we find that overall growth variability is condition-dependent, decreasing generally with mean growth rate. Our analysis shows that different limitations to growth, including nutrient uptake and catabolism, can result in the same growth phenotypes, underlining the robustness of the predicted behaviour. The sources of these fluctuations depend on the bottlenecks limiting growth. In complex environments, it is expected that several pathways limit growth and expression noise of several functional proteins will contribute to growth heterogeneity. Under combined nutrient uptake and catabolic limitations, for instance, both transporter and enzyme noise contribute to growth fluctuations (Supplementary Fig. 6), but components such as ribosomes that are not rate-limiting can be widely neglected (Fig. 3). Ribosome fluctuations become important in nutrient–drug interactions, when their inhibition by antibiotics imposes new limitations (Fig. 5), for which we predict complex responses that can either decrease or increase heterogeneity depending on nutrient conditions.

We analysed the propagation of stochastic fluctuations. We distinguished components upstream or downstream from growth, that is, cellular components transmitting fluctuations to growth or receiving them from growth, by building minimal delay graphs from pairwise cross-correlations. Species upstream of growth, such as transporter mRNAs and proteins as well as resources, correlate positively with growth rate, whereas species downstream

of growth either increase with growth or are diluted. These predictions may be tested using protocols similar to those employed in refs. [1,53]. We note, however, that cross-correlation analysis of macroscopic quantities does not identify the causal interactions mediated by individual biochemical reactions, and noise decompositions of cross-correlations[54] or recent methods from time-series analysis[55] could be useful to further investigate the propagation of stochastic fluctuations.

In particular, we identified resources to exhibit significant fluctuations, which they transmit to growth rate. In support of this, recent experiments showed that intracellular ATP levels can indeed vary substantially[47], and such variations can affect growth rate in eukaryotic cells[56]. Ribosome levels moreover correlate positively with growth rate, consistent with the known growth law[9]. But our model suggests that ribosome fluctuations follow those of growth rate, in agreement with the observation that asymmetric ribosome partitioning at cell division has negligible effect on growth rate[4]. Our finding that ribosome levels are set by growth rate at the single-cell level moreover suggests that growth fluctuations are a common source of cellular noise, and ribosomes propagate this noise to downstream processes[7,27,28].

The developed single-cell model allowed us to identify biological parameters that are otherwise non-identifiable using deterministic population-averaged approaches[35]. For example, total resource levels, as opposed to concentrations, have little to no effect on mean growth but affect growth rate variances, which can only be constrained by single-cell data (Supplementary Fig. 2). We further identified parameters such as transcription rates and negative feedback strength that can regulate growth heterogeneity (Supplementary Fig. 4). Recent work showed that growth fluctuations can impact the mean population growth[57], implying that these mechanistic parameters may be subject to evolutionary pressure. Cellular physiology could seize this degree of freedom and use it to shape noise to its benefit, for instance, as an evolutionary bet-hedging strategy[58–60]. The model could be put to use for in numero evolutionary experiments to test benefits of noise architectures and retrace possible evolutionary paths.

Our framework may also prove useful to benchmark the designs of synthetic circuits and increase their reliability. In this context one may, for example, wish to limit the impact of growth fluctuation on a circuit of interest[61]. Embedding such circuit in our model provides insights to re-architecture the global cellular noise to this effect. Finally, our results have important implications for drug tolerance and could pinpoint strategies to potentiate clinical treatment. Increased cell-to-cell variability can also drive phenotype switching. The latter plays a crucial role in persistence, a form of tolerance that allows bacteria to survive antibiotic treatment by switching to a dormant state[50], the precise mechanisms of which, however, are still unclear.

We limited our analysis to the effect of intrinsic fluctuations in the biochemical processes that dominate growth. We neglected variations in other processes linked to growth such as DNA replication, cell-cycle control[4], and production of structural components such as cell wall[62]. These may provide further contributions to growth heterogeneity, future investigations of which may yield salient insights into mechanisms behind tolerance to drugs that target these processes. We further focussed on the balance between catabolic and biosynthetic processes[9,63], where we considered effective regulation through the dependence of transcription on cellular resources (Methods, see also ref. [23]) and avoided mechanistic detail such as regulation via (p)ppGpp. In this sense, we mostly ignored specific regulatory processes such as involved in entering stationary phase, which may affect the quality of our predictions for poor growth media. Despite these limitations, our model recovers various types of data and empirical growth relations, highlighting the predictive power of our approach.

Our cell model links the stochasticity of intracellular mechanisms with growth variations observed in single cells and populations. Together with a theory to analyse stochastic cell and division dynamics, our work provides a framework to draw testable predictions and bring about a working understanding of the stochastic physiology in living cells.

## Methods

**List of reactions.** The model consists of stochastic reactions, adapted from a previous deterministic model[23], that represent transcription and degradation of mRNAs $m_y$, their binding to free ribosomes $p_r$ to form a ribosome-mRNA complex $c_y$, and translation reactions synthesising a protein $p_y$, where $y \in \{t, e, r, q\}$. To account for metabolism, we include uptake of an external nutrient $s$ at fixed concentration by a transport protein ($p_t$). The internalised nutrient $s_{\text{int}}$ is then catabolised to produce resource molecules $a$. In summary, the stoichiometries and propensities of the reactions are:

$$\emptyset \xrightarrow{\omega_y} m_y, \qquad m_y \xrightarrow{d_y} \emptyset, \tag{6}$$

$$p_r + m_y \underset{k_b}{\overset{k_u}{\rightleftharpoons}} c_y, \qquad n_y a + c_y \xrightarrow{\varsigma_y} p_r + m_y + p_y, \tag{7}$$

$$s \xrightarrow{v_{\text{imp}}} s_{\text{int}} \xrightarrow{v_{\text{cat}}} n_s a. \tag{8}$$

The propensities of mRNA degradation, ribosome binding and unbinding are modelled using mass action kinetics. Transcriptional and translational propensities depend on the resource $a$ and follow $\omega_y = M w_y \frac{X_a}{X_a + \theta_y}$ for $y \in \{r, e, t\}$, $\omega_q = M \frac{X_a}{X_a + \theta_q} \frac{w_q}{1 + (X_{pq}/K_q)^4}$ and $\varsigma_y = \frac{c_y \gamma_{\max} X_a}{n_y X_a + K_y}$ for $y \in \{t, e, r, q\}$, where $X_i = x_i/M$ denotes the concentration. Nutrient uptake and metabolism are modelled using quasi-steady state kinetics via the propensities: $v_{\text{imp}} = X_{p_t} v_t$ and $v_{\text{cat}} = X_{p_e} \frac{v_m X_{\text{sint}}}{K_m + X_{\text{sint}}}$. The instantaneous growth rate can be obtained in closed form using Eq. (3) and is a product of the translation elongation rate and the concentration of translating ribosomes

$$\lambda(\mathbf{X}) = \frac{\gamma_{\max} X_a}{X_a + K_\gamma} \sum_{y \in \{t,e,r,q\}} X_{c_y}, \tag{9}$$

assuming mass is dominated by protein content. In the model, mean growth rate is varied through nutrient quality $n_s$ describing the number of resource units produced per nutrient molecule.

To model operon architecture we replaced transcription, degradation, ribosome binding and translation reactions for transporters and enzyme species by the following set of reactions:

$$\emptyset \xrightarrow{\omega_e} m_{\text{bi}} \xrightarrow{d_e} \emptyset, \qquad p_r + m_{\text{bi}} \underset{k_b}{\overset{k_u}{\rightleftharpoons}} c_{\text{bi}}, \tag{10}$$

$$n_e a + c_{\text{bi}} \xrightarrow{\varsigma_e} m_{\text{bi}} + p_r + p_t + p_e$$

where $m_{\text{bi}}$ is the bicistronic mRNA species coding for both transporter and enzyme proteins, $p_t$ and $p_e$, respectively.

Chloramphenicol effectively reduces the pool of elongating ribosomes by binding to ribosomes and preventing elongation[64]. We model this effect using an additional reaction:

$$c_y \xrightarrow{k_{cm} X_{cm}} z_y,$$

describing ribosome inhibition by the imposed drug concentration $X_{cm}$ via a complex $z_y$ that is no longer available to translation.

**Small noise approximation.** In the limit of small fluctuations (large $M$), the mean concentrations $\overline{\mathbf{X}}$ and mean growth rate $\lambda(\overline{\mathbf{X}})$ are obtained by neglecting the noise terms in Eq. (4). In steady state, the balance between biomolecule synthesis and dilution due to cell growth determines these concentrations

$$\nu \mathbf{f}(\overline{\mathbf{X}}) = \lambda(\overline{\mathbf{X}}) \overline{\mathbf{X}}. \tag{11}$$

Mean cell mass $\overline{M}$ increases exponentially between divisions with deterministic time-intervals $\tau = \ln 2/\lambda(\overline{\mathbf{X}})$. Denoting by $O_c$ the concentration of origins at initiation, and by $\tau_{C+D}$ the time required to complete replication and trigger cell division (Fig. 1), the mean cell mass at birth and the number of origins are

exponential functions of the mean growth rate

$$M_0 = \frac{e^{\lambda(\overline{\mathbf{X}})\tau_{C+D}}}{2O_c}, O_0 = \frac{e^{\lambda(\overline{\mathbf{X}})\tau_{C+D}}}{2\ln 2}, \quad (12)$$

which follows from the delayed effect of initiation consistent with Donachie's result[25]. This implies a constant unit size (mass per number of origins) equal to ln $2/O_c$. To compare against bulk data, we used the relation $M_{\text{bulk}} = 2M_0 \ln 2$ (neglecting size variation before and after division[65]) and similarly for ribosome fractions and total mRNAs (Fig. 2a).

The small noise approximation allows computing the time-averaged covariance matrix $\overline{\Sigma} = \frac{1}{\tau}\int_0^\tau dt \, \text{Cov}[\mathbf{X}(t)]$ by solving the set of linear equations

$$0 = \mathcal{J}\overline{\Sigma} + \overline{\Sigma}\mathcal{J}^T + \frac{1}{2M_0\ln 2}\left(\lambda(\overline{\mathbf{X}})\Gamma + \sum_{r=1}^R \mathcal{D}_r\right), \quad (13)$$

where $\mathcal{J}(\overline{\mathbf{X}})$ is the Jacobian of the deterministic ODEs and $\Gamma(\overline{\mathbf{X}})$ and $\mathcal{D}_r(\overline{\mathbf{X}})$ are the noise strengths of cell divisions and of the biochemical reactions, respectively (see Supplementary Note 2 for details). The concentration covariances determine the size of growth rate fluctuations via

$$\text{CV}^2[\lambda] = \sum_{i,j=1}^N \frac{\partial \ln\lambda(\overline{\mathbf{X}})}{\partial \ln \overline{X}_i} \frac{\overline{\Sigma}_{ij}}{\overline{X}_i \overline{X}_j} \frac{\partial \ln\lambda(\overline{\mathbf{X}})}{\partial \ln \overline{X}_j}. \quad (14)$$

We analyse the sources of growth variations by decomposing Eqs. (13) and (14) into contributions of cell divisions or groups of reactions (Supplementary Note 2).

**Model parametrisation.** We parametrised the model with literature values for *E. coli* (Supplementary Table 1). We estimated the transcription rates and the scaling of transcription and translation rates with resource levels using the small noise approximation, Eqs. (11)–(14), combined with MCMC parameter sampling (Supplementary Fig. 1, see Supplementary Note 3). To constrain these parameters, we used bulk data of mean ribosomal mass fraction against mean growth rate[9] and the dependence of the $\text{CV}^2[\lambda]$ on the mean from two recently published data sets of single-cell time-lapse microscopy[1,3] covering a broad range of growth rates. We further constrained the maximal growth rate to 3.75 dbl/hr (16 min doubling time).

**Stochastic simulations.** We use a hybrid scheme that simulates reactions either using the next-reaction method or ODEs[66]. To account for non-exponential reaction-time distributions, we update propensities every 0.05 min. Supported by the predictions using the small noise approximation, it was sufficient to simulate only those reactions stochastically that change the lowly abundant mRNAs of transporters and enzymes and their corresponding ribosomal complexes. We determine growth rate statistics using Eq. (3).

**Code availability.** All codes are available from the authors. An extensible Mathematica Notebook performing the small noise approximation and noise decomposition is available on figshare[68].

## Data availability

All relevant data are available from the authors.

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

## Acknowledgements

We thank Mauricio Barahona, Meriem El Karoui, Diego Oyarzún and Peter Swain for valuable feedback. P.T. gratefully acknowledges support from The Royal Commission for the Exhibition of 1851 and the EPSRC Centre for Mathematics of Precision Healthcare (EP/N014529/1), and V.D., A.Y.W. and G.T. from the ERC's RULE project. A.Y.W. is affiliated with the National Institute for Health Research Health Protection Research Unit (NIHR HPRU) in Healthcare Associated Infection/Antimicrobial Resistance at Imperial College London in partnership with Public Health England (PHE). The views expressed are those of the author(s) and are not necessarily those of the National Health Service (NHS), the NIHR, the Department of Health or PHE.

## Author contributions

P.T., V.D. and A.Y.W. designed research; P.T., G.T., V.D. and A.Y.W. performed research; P.T. and A.Y.W. analysed data and wrote the paper.

## Additional information

**Competing interests:** The authors declare no competing interests.

