## [Peer Review File · Nature Communications]

Reviewers' comments:

Reviewer #1 (Remarks to the Author):

In this manuscript, the authors present a stochastic coarse-grained whole-cell model designed to describe the mechanisms and origins of growth-rate fluctuations in bacterial cells. The model integrates minimal models of metabolism, gene expression, growth, cell division and cell-size regulation, partly based on a deterministic model published earlier by the last author.

To analyze this model, the authors formulate a neat stochastic reaction-division framework based on Langevin equations that is applicable to the model above, but can also be used for other models with growth and division. In a small-noise approximation, the time averages and time-averaged covariances of the relevant variables can be calculated analytically; in addition, the relative importance of the various noise sources for the growth-rate variations can be quantified.

Using the above techniques, the whole-cell model above is fit to several experimental data sets, after which the properties of the fitted model are explored and compared to various experimental observations.

Generally, the paper is well written and the story well presented. The computational and mathematical techniques developed and employed are innovative, advanced, and generally sound. The authors correctly note that this is one of the first attempts to combine recent ideas on the global organization of bacterial cells with stochastic modeling. I've learned a lot studying the paper and its supplement.

That said, I do have a few questions/doubts, which I hope the authors can clarify/address.

* It seems to me that some of the results could be sensitive to the parameters chosen for the model. While the authors briefly discuss the procedures used for estimating the parameters in both main text and supplement and list the parameter values used in Table S1, I still had trouble understanding the procedure. I hope that the authors can expand (one of) these sections. In particular:

-Which exact data from refs. 1, 2, and 8 haven been used?

-How were these data used? In particular, in ref. 1 growth rate was varied by titrating the mean expression of the lac operon, which I suppose amounts to changing parameter $\$w_t\$$ and possibly $\$w_e\$$ whereas other studies vary the growth medium, which presumably affects $\$v_t\$$ and/or $\$n_s\$$. Where all data interpreted as if the only parameter that changes between conditions is $\$n_s\$$?

-What is the parameter $\$zeta\$$ (called the (effective) resource level(s)) mentioned in Fig. S1, S2, Table S1, and in line 288 of the supplement? It may just be my, but don't see it defined anywhere.

-Are the results sensitive to changes in the model parameters?

* In the section "Condition-dependence of growth rate fluctuations" it is stated that the CV reduces to 0 as the mean growth rate approaches its maximum. From Fig. 3 I can see that this is clearly true, but I do not understand it. From equation (6) I conclude that, even at saturating $\$X_a\$$ the growth rate should still be affected by fluctuations in the ribosome concentration. I must be missing something. What?

* It seems that at high import rates the long-time average of $\$s_{int}\$$ and $\$a\$$ may not exist (they may increase indefinitely). Is that right? Does that affect the results?

* The cross-correlation analysis is very interesting, but seems complicated by the old problem that correlation does not imply causation. How can you distinguish, for instance, a short delay between A and B due to a direct causal effect from a short delay because both respond to common cause C

with a slightly different delay?

* Case in point: Based on the cross-correlation plot of Fig. 4A, it is stated that growth rate affects ribosome concentration with a delay. What is the mechanism causing this delay? Can the authors exclude that growth and ribosome concentration respond to the same signal with a different delay rather than affecting each other?

* The authors write:

"The model does not comprise effects that are detrimental to growth. Any upstream components, that is, species transmitting fluctuations to growth, are therefore expected to promote growth and so correlate positively with it."

Again, the results seem consistent with this statement, but I don't understand it fundamentally. If I understand well, any fluctuation in the concentration one molecule must be compensated by a fluctuation in another one because the cell density is presumed fixed (Equation S6). Therefore a fluctuation in a component that is not limiting may systematically come at the expense of a more limiting component, which would result in a negative correlation with the growth rate. What excludes this possibility?

Minor points:

* The model and all data presented are based on E. coli. I suggest that the authors at least mention this in the introduction; most information presented there is also based on E. coli but the species is never mentioned.

* The details of the biological part of the model are relegated to the methods section at the end of the paper. This makes it very hard to interpret the results without first studying the methods section in detail. For example, in several figures (including Fig. 2 and 3), the mean growth rate is varied (x-axis). It took me a while to find, in the methods section, just how the growth rate is varied in the model (through variation of n_s), which I felt was essential. For a reader who has not read the models section, Fig. 3 is somewhat mysterious, because the parameters v_t and v_m have not been introduced at this point, and it is unclear why their ratio v_t/v_m stays fixed while the growth rate is varied, presumably by changing the nutrient. I would suggest to include more information about the model to the results section.

* In the caption of Fig 3, I believe $v_t > v_m$ should be $v_t < v_m$ and vice versa.

* Page 9, last line of second column: should p_t be X_{p_t} and p_m be X_{p_e} ?

* Table 1: Parameter h_q does not occur in the model (is replaced by its value 4 on page 9).

Reviewer #2 (Remarks to the Author):

In this paper, the authors extend an existing model with the aim to predict cellular phenotypes related to growth, most notably metabolic heterogeneity. These phenotypes are important, novel, and topical, with recent experimental efforts along similar lines.

I do still have some issues that would need to be clarified before I can fully judge the importance of the work. My main impression though is that this is a very nice piece of work. It is a very important aim to try to understand what the causes of cellular and metabolic heterogeneity are,

and the quite minimal yet dynamic approach that is followed here is one that is required within the field. The work appears well thought through, makes the right choices, and provides a large number of interesting observations / predictions.

Let me first try convey what is my most overarching concern. The study presents a model that appears to make predictions that are consistent with a number of experimental observations. The model contains a number of underlying details on molecular composition, which feed the novel conclusions of this work. All of these findings rest on premise that this one model correctly captures the main mechanisms, and hence on the comparison with the experimental data, which is mainly at the higher macroscopic level. While the comparisons look ok, it is just one model. No other models are compared, nor is it made clear why this particular model, or which aspects of it, make it so that the predictions appear correct. As mentioned a contrast with other models, or variants of the model, can help to pinpoint what the key elements are. This is important, because quite far reaching and detailed conclusions are drawn, and we/I need to be convinced that they are correct. In a sense the work is lacking in a key property of scientific study, namely that it must be falsifiable. This criticism requires not a minor explanation, but a solid effort, also in the paper.

I note that the above is for instance relevant (though not exclusively) to the key Fig. 2e. Why are the predictions correct both quantitatively and in terms of the trend? Which models or changes to the model would give something different? Given the generality of the model, should those same noise levels then not be identical in different strains and or even different types of bacteria? I do not think this is the case, or is it? Or did I miss something and is some fitting done here to make the theory match the data?

More specifically:

1) Related to the point above, it would help if the text is far more explicit about whether any fitting is done (if at all), and what information the choice of parameters is based on. This is essential because for predictions one can only use prior information, not the information that is being predicted, obviously. There is some clarity already. On p. 4 it is stated that the first two statistical moments are inferred. If this means this is how the noise intensities were determined, then this should be discussed in such words so that biologists can also understand it, as Intensity of any noise sources are central to the story. But I also find this to brief in scientific terms. The authors should provide evidence that intensities of noise sources inferred this way are actually correct, and match with experimental data. I understand there is some evidence in Fig. 2e, but this is the overall growth noise, not the sources of noise. Then on page 8, I read 'without refitting', which suggests that fitting was performed before. If so this should be described, as my general understanding was that nothing was fitted. But more generally, it should be made more transparent what is assumed, which independent data is used, and how parameters were chosen.

2) The paper discussed cross correlations that were measured by Kiviet et al., which is quite interesting. It would also be interesting to quantify the autocorrelation functions and related relaxation times, as done in the Kiviet paper. It might provide a hypothesis about their observed shorter relaxation time for growth than for expression.

3) The model ignores any regulation. Perhaps regulation is not required to explain the data, but it would be good to comment on this, and/or test how known regulatory interactions would affect the result.

4) Some conclusions are presented as results, but they have already been established in previous work, which should be described and cited. For instance [We moreover find that growth rate is a source of global noise that can be transmitted to other processes, for example, via ribosomes^{6,26,27}] was one of the central conclusions of the Kiviet paper. Similarly, this paper concluded similar things as [This is not because intracellular reactions stop fluctuating, but rather because growth rate saturates and thus such fluctuations no longer translate to growth

variability.]

5) One element of the model is that [Cellular protein content dominates biomass, and thus the total translation rate determines the rate of biomass production^{8,22}]. It implies that the rate of growth is equal to the total translation rate. This is stated as a certainty but has in fact not been shown in a dynamical context, as cell wall and volume could exhibit fluctuations that are not directly dependent on translation. It may be true in steady state, but could very well be different dynamically. Hence it is an assumption, and should be described as such, argued for, and discussed in terms of limitations. Moreover, this assumption appears directly linked to one of the main conclusion, which is that transcription dominates growth noise. This appears expected in a way given this assumption.

6) Another assumption of the model is also the various simplifications that are used, such as the focus on one transporter and enzyme. Simplifying is fine, but one should argue for this choice, and limitations discussed. It would also be good to test how relaxing this assumption changes the results. Obviously this is a major simplification, as there are many and many types of genes contributing to growth.

7) On page 5, it reads 'very small growth rates'. Please be specific here and in other such instances.

Response to comments by reviewer #1:

1) It seems to me that some of the results could be sensitive to the parameters chosen for the model. While the authors briefly discuss the procedures used for estimating the parameters in both main text and supplement and list the parameter values used in Table S1, I still had trouble understanding the procedure. I hope that the authors can expand (one of) these sections. In particular:

- A. Which exact data from refs. 1, 2, and 8 haven been used?
- B. How were these data used? In particular, in ref. 1 growth rate was varied by titrating the mean expression of the lac operon, which I suppose amounts to changing parameter w_t and possibly w_e whereas other studies vary the growth medium, which presumably affects v_t and/or n_s . Where all data interpreted as if the only parameter that changes between conditions is n_s ?
- C. What is the parameter ζ (called the (effective) resource level(s)) mentioned in Fig. S1, S2, Table S1, and in line 288 of the supplement? It may just be my, but don't see it defined anywhere.
- D. Are the results sensitive to changes in the model parameters?

This is an important comment. We now expand on the procedure used for parameter estimation in the main text and clearly reference where to find further details in the supplementary material. We also provide a new analysis regarding the sensitivity of growth rate and its fluctuations to all model parameters.

More specifically, regarding points A-D:

- A. Previously, we included information on the data used for model fitting in Fig. 2 (filled markers). In the revised manuscript, we now use more prominent colours to highlight data used for fitting and validation. In addition, we detail the data used for the fitting in the main text (line 224-32), namely (i) ribosomal mass fractions at different growth rates were fitted against data from Ref 8, and (ii) growth rate CV's in various growth conditions from Refs 1& 3, and (iii) and imposed maximal growth rate of 3.75 doubling per hour. Technical details of the fitting can be found in the Methods Section c and SI Sec. III.
- B. We vary only the parameter n_s across different conditions. As correctly mentioned by the reviewer, this interpretation applies to the data by Kennard et al. (Ref 3) but requires additional assumptions when comparing with the data by Kiviet et al. (Ref 1). Via sensitivity analysis, we now show that both w_t and v_t have qualitatively similar effects on the model behaviour as n_s . Namely growth rate decreases and its fluctuations increases when decreasing any of these parameters as mentioned in the caption of Fig. 2a and summarised in Fig. S4.
- C. Apologies for the missing definition. The parameter ζ scales the Michaelis-Menten constants of translation and transcription and thus effectively determines the levels of intracellular resources. Since the definition is not a kinetic parameter but rather affects the rate constants, we explained its meaning in the SI (lines 287-290), in the caption of Table I in the SI and added the necessary references to SI Fig. S2.

"Stochasticity of cellular growth: sources, propagation and consequences"

D. This is an interesting point, which we did not fully address in the previous manuscript. We therefore carried out an extensive sensitivity analysis of mean growth rate and its fluctuations to *all* model parameters. The results are summarised in the new Fig S4, and we included an extensive discussion of the implications of these sensitivities to physiologically relevant parameters and their biological interpretation. We moreover highlight that parameter sensitivity can be used to assess the potential of regulating growth rate fluctuations (see also comment 12 by reviewer #2). We included a detailed discussion of these effects in the main text at the end of Sec. B.

2) In the section "Condition-dependence of growth rate fluctuations" it is stated that the CV reduces to 0 as the mean growth rate approaches its maximum. From Fig. 3 I can see that this is clearly true, but I do not understand it. From equation (6) I conclude that, even at saturating X_a the growth rate should still be affected by fluctuations in the ribosome concentration. I must be missing something. What?

The reviewer is right, fluctuations in ribosomes would certainly affect growth rate. However, with the growth limitations considered previously (uptake & catabolism), ribosomes are highly abundant, especially at fast growth rates, and so expression of ribosomes transmits only negligible fluctuations to growth.

To assess the potential impact of ribosome fluctuations, we now explore an alternative scenario where ribosomes essentially limit growth. The results are shown in the new Fig. 3. Under these limitations, ribosome fluctuations contribute noise also at high growth rates, but the scenario requires unrealistically small numbers of ribosomal transcripts (see also point 10). We modified the paragraph (lines 279-89) to better explain the implicit assumptions of the phenomenon.

3) It seems that at high import rates the long-time average of s_{int} and a may not exist (they may increase indefinitely). Is that right? Does that affect the results?

This concern does not apply because intracellular concentrations are diluted by cell divisions as shown in Eq. (4a) and in Eq. (7) for the corresponding averages. Any increase in production rate will eventually be compensated by dilution and thus s_{int} and a both attain stable concentrations.

In the example mentioned by the reviewer, the rate of change of intracellular concentration $X_{s_{int}}$ of s_{int} is

$$\frac{\partial X_{s_{int}}}{\partial t} = v_{imp} - \frac{v_m X_{pe} X_{s_{int}}}{K_m + X_{s_{int}}} - \lambda X_{s_{int}},$$

which always admits a positive solution if the growth rate λ is nonzero. Even if $v_{imp} \gg v_m X_{pe}$, any import rate will eventually be outweighed by dilution once $X_{s_{int}}$

"Stochasticity of cellular growth: sources, propagation and consequences"

becomes sufficiently large, as we illustrate in the figure below. The same arguments hold for the concentration of ϕ .

Fig: Example of high import rate. For sufficiently large s_{int} dilution dominates, the intersection of the two lines yields a positive steady state.

4) The cross-correlation analysis is very interesting, but seems complicated by the old problem that correlation does not imply causation. How can you distinguish, for instance, a short delay between A and B due to a direct causal effect from a short delay because both respond to common cause C with a slightly different delay?

We thank the reviewer for bringing this important point to our attention. Previously, we did not distinguish the observed correlations clearly enough from causal interactions.

In our model, causal effects are implemented by specific biochemical reactions, which mediate interactions between the molecules of the different species. The interactions between these species can be quantified for instance using the Jacobian matrix, which can be derived analytically from the rate equations (7). We quantified the effects of these interactions on growth by deriving an explicit analytical formula for the growth rate, Eq. (6). The formula depends on the concentrations of resources and the concentration of translating ribosomes. This implies that all of these species have a direct causal effect on the growth rate.

In the revised manuscript, we explain that the cross-correlations analysis focuses on the temporal order among fluctuations in macroscopically measurable quantities (lines 416-32). To clarify, our model consists of a number of microscopic species that interact via biochemical reactions, which are causal. But in the analysis we assume that a smaller number of lumped macroscopic concentrations, such as total ribosome concentrations, are observed. The correlations between these macroscopic variables and growth rate are difficult to predict from first principles because the reduced variables involve spurious correlations, such as those noted by the reviewer.

- A. Case in point: Based on the cross-correlation plot of Fig. 4A, it is stated that growth rate affects ribosome concentration with a delay. What is the mechanism causing this delay? Can the authors exclude that growth and ribosome concentration respond to the same signal with a different delay rather than affecting each other?

"Stochasticity of cellular growth: sources, propagation and consequences"

We agree that it is difficult to distinguish these interactions using a correlation-based analysis. Consider, for instance, the correlation between resources, growth rate and total ribosome concentrations. In this case, we find that resources transmit fluctuations to growth rate, which transmit them to ribosomes. However, growth rate and transcription of ribosomes increase with resource levels. It is therefore plausible that resources affect ribosomes and growth but with a different delay, resulting in the observed spurious correlation as noted by the reviewer.

On the other hand, an increase in growth rate correlates with increased overall transcription. Since ribosomes are produced autocatalytically such an increase in transcription could lead to an overproportional translation of ribosomes, which is not compensated by dilution. This scenario is also compatible with the observed correlations.

Both scenarios, the one described by the reviewer in A and the one mentioned in the paragraph above, are plausible, and we cannot exclude either possibility. In particular, our analysis does not exclude further correlations but only reports the dominant ones. We expect both scenarios to be relevant in practice.

This can also be seen from our cross-correlation analysis in different growth conditions (SI Fig. S8a). In slow and moderate growth conditions r-mRNA concentrations are downstream of growth rate, which is compatible with the second scenario, while they are upstream of growth rate in fast growth conditions, compatible with the first scenario mentioned by the reviewer (SI Fig. S8b). This suggests that both effects are relevant for the dynamics but their relative contribution to the observed correlations changes across growth conditions.

New methods from time-series analysis that exploit more general concepts of statistical dependence could be useful to further characterise the causal underpinnings of correlations (Ref. 54). But such methods are the subject of ongoing research and, in our experience, rely on additional parameters, which can affect conclusions drawn from the analysis. As explained above, several effects plausibly contribute to the dynamics of the model, and so even elaborate methods would not be able to exclude either effect. In the discussion (lines 623-8), we now highlight that new techniques in time-series analysis can provide future insights.

- B. The authors write: "The model does not comprise effects that are detrimental to growth. Any upstream components, that is, species transmitting fluctuations to growth, are therefore expected to promote growth and so correlate positively with it."

Again, the results seem consistent with this statement, but I don't understand it fundamentally. If I understand well, any fluctuation in the concentration one molecule must be compensated by a fluctuation in another one because the cell density is presumed fixed (Equation S6). Therefore a fluctuation in a component that is not limiting may systematically come at the expense of a more limiting component, which would result in a negative correlation with the growth rate. What excludes this possibility?

The reviewer is right, the statement is confusing, and we removed it from the revised manuscript. Fluctuations in non-limiting components can indeed affect other components, which is taken into account via a dilution term in Eq. (4a). For example, we observe that fluctuations in ribosomes, which are non-limiting in the cases considered in Fig 4c, correlate negatively with several other protein species. However, the observation that upstream components such as transporter proteins correlate positively with growth rate means that their positive effect on growth dominates over the transmission of ribosome fluctuations via a fixed cell density.

To clarify, the delay graphs in 4c compress information from the mutual cross-correlation functions (4a) - they only visualise maximal correlations between species with minimal delay and ignore other correlations. For example, the cross-correlation functions shown in Fig. 4a display a spectrum of positive and negative correlations over a range of positive and negative lags (SI Fig. S8b), some of which may be due to dilution effects as mentioned by the reviewer. We included a note on the possibility of other correlations not visualised in the minimal delay graphs (lines 456-9).

Minor points:

5) The model and all data presented are based on *E. coli*. I suggest that the authors at least mention this in the introduction; most information presented there is also based on *E. coli* but the species is never mentioned.

Apologies, we now included a note clarifying this in the introduction (line 78).

6) The details of the biological part of the model are relegated to the methods section at the end of the paper. This makes it very hard to interpret the results without first studying the methods section in detail. For example, in several figures (including Fig. 2 and 3), the mean growth rate is varied (x-axis). It took me a while to find, in the methods section, just how the growth rate is varied in the model (through variation of ν_n), which I felt was essential. For a reader who has not read the models section, Fig. 3 is somewhat mysterious, because the parameters ν_t and ν_m have not been introduced at this point, and it is unclear why their ratio ν_t/ν_m stays fixed while the growth rate is varied, presumably by changing the nutrient. I would suggest to include more information about the model to the results section.

In the caption of Fig. 2, we now explain how we varied growth rate (see also response to point 1). We give an overview of the key biological components of the model in the first four paragraphs of Sec. IIA and Fig. 1, but due to length restrictions we decided to relegate the relevant equations to the Methods section.

Apologies, we now removed any references to ν_t and ν_m from the main text. To make the presentation accessible to a broad audience we now include an illustration of the dominant reaction fluxes in Fig. 3.

7) In the caption of Fig 3, I believe $\nu_t > \nu_m$ should be $\nu_t < \nu_m$ and vice versa.

The reviewer is right, apologies for the confusion - we fixed this in the revised manuscript.

8) Page 9, last line of second column: should p_t be X_{p_t} and p_m be X_{p_e} ?

The reviewer is right - we changed it.

9) Table 1: Parameter h_q does not occur in the model (is replaced by its value 4 on page 9).

Apologies, we removed this parameter from the table.

Response to comments by reviewer #2:

10) Let me first try convey what is my most overarching concern. The study presents a model that appears to make predictions that are consistent with a number of experimental observations. The model contains a number of underlying details on molecular composition, which feed the novel conclusions of this work. All of these findings rest on premise that this one model correctly captures the main mechanisms, and hence on the comparison with the experimental data, which is mainly at the higher macroscopic level. While the comparisons look ok, it is just one model. No other models are compared, nor is it made clear why this particular model, or which aspects of it, make it so that the predictions appear correct. As mentioned a contrast with other models, or variants of the model, can help to pinpoint what the key elements are. This is important, because quite far reaching and detailed conclusions are drawn, and we/I need to be convinced that they are correct. In a sense the work is lacking in a key property of scientific study, namely that it must be falsifiable. This criticism requires not a minor explanation, but a solid effort, also in the paper.

We went through substantial efforts to address this concern. We added further explanation, carried out additional analyses (described in Sec. IIC) and included additional figures (Fig. 3, S7) as described below.

We agree that many models can fit the data. In our previous submission we showed that both a model with limitations in nutrient uptake and a model with limitations in catabolism is consistent with the data. Under these respective limitations, either expression of transporter or enzyme mRNAs explain the observed levels of growth fluctuations. Within the previous analysis, however, we could not exclude the potential impact of fluctuations in the expression of ribosomes on growth variations.

To pinpoint key elements responsible for growth heterogeneity, we now include a new analysis in the revised manuscript, we completely rewrote Sec. IIC '*Limiting factors to growth*', updated Fig. 3 with a new panel, and we included a new supplementary figure (Fig. S7). More specifically, we formulate an alternative model, which considers a constant influx

"Stochasticity of cellular growth: sources, propagation and consequences"

of resources and lacks expression of metabolic components. It tests the hypothesis whether ribosome expression can act as a key source of growth fluctuations.

To explain the observed levels of fluctuations, the alternative model suggests that ribosomal messengers are present at low copy numbers. To constrain our model further, we compared the predicted r-mRNA numbers with literature values (Bernstein, 2002), which are several orders of magnitudes higher (Fig S7f). In contrast, values predicted under limitations in nutrient uptake or catabolism are in excellent quantitative agreement with the data (Fig S7f).

Altogether, we analysed and compared various potential growth limitations, which are now summarised in the updated Fig. 3. The results suggest that expression of ribosomes (or q-protein) is not a major source of fluctuations in growth rate (lines 334-55). In contrast, the corresponding abundances of transporter and enzyme mRNAs represent plausible sources of growth variations, which we discuss in Sec. IIC (lines 356-73). This provides additional support for our finding that expression of metabolic components constitutes a key source of cellular noise - among other sources as discussed below in point 14).

We further analysed the effects of parameter variations on our model (Sec. IIC lines 373-406, SI Fig. S4), which are described in the response to point 1D of reviewer #1 and point 12 below.

10a) Related to the point above, it would help if the text is far more explicit about whether any fitting is done (if at all), and what information the choice of parameters is based on. This is essential because for predictions one can only use prior information, not the information that is being predicted, obviously. There is some clarity already. On p. 4 it is stated that the first two statistical moments are inferred. If this means this is how the noise intensities were determined, then this should be discussed in such words so that biologists can also understand it, as Intensity of any noise sources are central to the story. But I also find this too brief in scientific terms. The authors should provide evidence that intensities of noise sources inferred this way are actually correct, and match with experimental data. I understand there is some evidence in Fig. 2e, but this is the overall growth noise, not the sources of noise. Then on page 8, I read 'without refitting', which suggests that fitting was performed before. If so this should be described, as my general understanding was that nothing was fitted. But more generally, it should be made more transparent what is assumed, which independent data is used, and how parameters were chosen.

We apologise for the confusion. As mentioned in our response to comment 1) by reviewer #1, we now expand on the procedure employed for parameter estimation and include details on the data used for estimation and validation in Fig. 2 and in the main text.

We identified transcription, degradation and partitioning of transporter mRNAs as the dominant noise sources. The noise intensities of these processes are derived from the reaction equations given in Methods Sec. a, which in turn depend on kinetic parameters. The relevant dilution rate is given by the growth rate (see Eq. 4a), the mRNA degradation was taken from literature values (SI Table 1), and the relevant transcription rates have been inferred from the data as we detail now in Methods Sec. c.

"Stochasticity of cellular growth: sources, propagation and consequences"

To provide evidence that the inferred sources of growth fluctuations are actually correct, we now compare predicted mRNA abundances to literature values. Predicted abundances of enzymatic and transporter transcripts are consistent with values for essential genes. We also compare the predicted numbers of ribosomal transcripts with experimental data and find good agreement across two growth conditions (SI Fig. S7).

We further explore the hypothesis that growth variability could be explained by alternative noise sources (Fig. 3c), but find that it is not supported by experiments (SI Fig. S7). Instead, the data support our conclusion that the expression of metabolic components is a major source of growth rate fluctuations. To detail these findings, we completely rewrote the paragraphs on *'Limitations to growth'* (lines 334-371).

11) The paper discussed cross correlations that were measured by Kiviet et al., which is quite interesting. It would also be interesting to quantify the autocorrelation functions and related relaxation times, as done in the Kiviet paper. It might provide a hypothesis about their observed shorter relaxation time for growth than for expression.

Thanks very much for this interesting suggestion. We analysed the autocorrelation times of fluctuations in growth rate and transporter proteins, see figure below. The predicted correlation times are quantitatively comparable, in agreement with the findings in Kiviet et al. In particular, they support the conclusion that fluctuations in gene expression propagate to growth rate. Besides this, the analysis did not provide further insights, which is why we did not include it in the revised manuscript.

Figure: In our model, autocorrelation times in growth rate scale with autocorrelation times in transporter proteins.

The observation mentioned by the reviewer, that growth rate exhibits a smaller autocorrelation time than protein levels, was not central to Kiviet's study. It could suggest that expression of metabolic components is not the only source of growth fluctuations, and that other unconsidered processes inject extrinsic noise that decorrelate growth and protein expression. We added a note on these limitations in the discussion (line 677-85).

12) The model ignores any regulation. Perhaps regulation is not required to explain the data, but it would be good to comment on this, and/or test how known regulatory interactions would affect the result.

We thank the reviewer for this interesting suggestion. It inspired a new analysis and we devoted a whole new paragraph in Sec. IIC on *'Potential regulation of growth fluctuations'*. To address this question, we exploit that the effect of regulatory interactions can be interpreted as effective growth-rate dependencies of parameters. For example, transcription rates may depend on the abundance of a specific transcription factor, whose level varies with growth rate. To explore such effects, we now analyse the sensitivity of mean growth rate and its fluctuations to all model parameters in various growth conditions (see also comment 1 by reviewer #1).

The new sensitivity analysis allows us to identify potential targets for regulation of growth heterogeneity. For example, it suggests that regulation of DNA replication can have a profound effect on heterogeneity because it affects cells size, which in turn scales overall noise levels. It further shows that some known regulatory interactions such as downregulation of ribosomal expression (e.g. via ppGpp), upregulation of enzyme expression (e.g. via cAMP) and negative autoregulation - a ubiquitous network motif found across bacterial gene regulation - can attenuate growth heterogeneity. The results therefore raise interesting questions regarding evolutionary benefits of these regulatory interactions, on which we briefly elaborate in the discussion (lines 650-62).

13) Some conclusions are presented as results, but they have already been established in previous work, which should be described and cited. For instance [We moreover find that growth rate is a source of global noise that can be transmitted to other processes, for example, via ribosomes^{6,26,27}] was one of the central conclusions of the Kiviet paper. Similarly, this paper concluded similar things as [This is not because intracellular reactions stop fluctuating, but rather because growth rate saturates and thus such fluctuations no longer translate to growth variability.]

We thank the reviewer for this suggestion and we now worded the corresponding paragraphs (lines 99-101 and 280-289) more carefully. For instance, we point out that Kiviet et al. concluded that growth rate represents a source of global noise. Here we identify that growth fluctuations may be transmitted via ribosomes. We further elaborate on how transmission of molecular fluctuations to growth rate can be interpreted within our model assumptions, especially at high growth rates not considered by Kiviet et al.

14) One element of the model is that [Cellular protein content dominates biomass, and thus the total translation rate determines the rate of biomass production^{8,22}]. It implies that the rate of growth is equal to the total translation rate. This is stated as a certainty but has in fact not been shown in a dynamical context, as cell wall and volume could exhibit fluctuations that are not directly dependent on translation. It may be true in steady state, but could very well be different dynamically. Hence it is an assumption, and should be described as such, argued for, and discussed in terms of limitations. Moreover, this assumption appears

"Stochasticity of cellular growth: sources, propagation and consequences"

directly linked to one of the main conclusion, which is that transcription dominates growth noise. This appears expected in a way given this assumption.

Apologies, we now clearly indicate and motivate the link between translation and growth rate as an assumption (line 115-7), and we discuss its limitations in the discussion (line 677-85).

That transcriptional noise can act as a source of cellular heterogeneity is in agreement with many studies of cellular noise and thus indeed not unexpected. One of our main conclusions, however, is that transcription of metabolic components constitutes a key source of growth variations. This was first suggested by Kiviet et al., and our work provides the first theoretical analysis of this phenomenon that quantifies its overall contribution to growth noise.

Our analysis in Sec. IIC '*Processes contributing to growth fluctuations*' suggests that the contribution of transcription is significant but it is not the only source. Specifically, transcription contributes only half of the noise in growth rate consistently across different growth conditions. The other half originates from the removal of metabolic mRNAs via degradation and, particularly, their partitioning at cell division (Fig 2e and Fig 3). This contribution of partitioning noise is robust across all limitation models considered and has not been previously studied. Following this comment, we now highlight the role of partitioning noise more clearly in the respective paragraph (lines 295-302).

15) Another assumption of the model is also the various simplifications that are used, such as the focus on one transporter and enzyme. Simplifying is fine, but one should argue for this choice, and limitations discussed. It would also be good to test how relaxing this assumption changes the results. Obviously this is a major simplification, as there are many and many types of genes contributing to growth.

We apologise that this assumption was not clear in the previous manuscript. We now explain that we focus on the situation where the transporter and enzyme species represent metabolic bottlenecks (lines 134-135). Under this assumption, we find that the components limiting growth will be the major factors contributing to growth heterogeneity. On the other hand, dynamics of components which do not limit growth, such as enzymes and ribosomes that are not rate-limiting, will not significantly contribute to fluctuations in growth rate (Fig. 3).

We agree that the general situation is more complex since one needs to single out all pathways limiting growth. We now emphasize that when several bottlenecks exist, all of these processes will contribute to growth heterogeneity (line 325-33, SI Fig. S6). For example, when nutrient uptake and catabolism limit growth both enzyme and transporters contribute to growth heterogeneity. In nutrient-drug environments, antibiotics inhibit ribosome expression and so ribosome expression becomes rate-limiting. In these conditions we predict complex patterns that either decrease or increase heterogeneity. We now also elaborate on these aspects in Sec. IIE (lines 516-22) and in the discussion (line 592-611).

16) On page 5, it reads 'very small growth rates'. Please be specific here and in other such instances.

"Stochasticity of cellular growth: sources, propagation and consequences"

Apologies, we now specify growth regimes clearly in Fig. S4, S5, S8 and throughout the main text (see caption of Fig. 4, and lines 299-302, 308).

REVIEWERS' COMMENTS:

Reviewer #1 (Remarks to the Author):

In response to the issues raised by both reviewers, the authors have made several significant changes to the manuscript. Overall, I feel that most of the concerns have been addressed. In particular:

In response to my comment 1) the authors expanded their explanation of the fitting procedure used, which is more clear now. They also included a sensitivity analysis, which partly addresses the question of how sensitive the results are with respect to parameter variation.

In response to comment 2) the authors rewrote the corresponding paragraph; this removed my objections.

My comment 3) was based on a mistake of my own; I should have realized that this model includes dilution of metabolites by growth, as the authors explained.

Comment 4A/B): The authors clearly seem to agree with my point that correlations between A and B do not necessarily represent causal interactions between them and that a longer delay between A and C than between B and C does not necessarily imply that information flows from A to C via B. They also now write about this in the manuscript, which is good. And to be clear, I do like the idea of the analysis as well as the technical implementation. Nevertheless, the choice of words in the manuscript still suggests causal mechanisms that, as I see it, cannot be inferred with certainty from the correlations. For instance, the caption of Fig. 4 says that if B has a minimal delay from A, then B is the first to receive fluctuations from species A. I don't think this is necessarily true, because it could be that A does not "talk" to B at all; both could receive fluctuations from some variable Z at a different delay. To solve this objection, it is probably sufficient to "soften" a few sentences in several places; for instance, "which suggests that species B is the first to receive fluctuations from species A" rather than "so species B are the first to receive fluctuations from species A".

Minor comments (5 -- 9): the authors fixed these issues.

Reviewer #2 (Remarks to the Author):

The authors have done solid work in this this review. My central concerns are now addressed, and I support publication.

Comments by Reviewer #1:

In response to the issues raised by both reviewers, the authors have made several significant changes to the manuscript. Overall, I feel that most of the concerns have been addressed. In particular:

In response to my comment 1) the authors expanded their explanation of the fitting procedure used, which is more clear now. They also included a sensitivity analysis, which partly addresses the question of how sensitive the results are with respect to parameter variation.

In response to comment 2) the authors rewrote the corresponding paragraph; this removed my objections.

My comment 3) was based on a mistake of my own; I should have realized that this model includes dilution of metabolites by growth, as the authors explained.

Comment 4A/B): The authors clearly seem to agree with my point that correlations between A and B do not necessarily represent causal interactions between them and that a longer delay between A and C than between B and C does not necessarily imply that information flows from A to C via B. They also now write about this in the manuscript, which is good. And to be clear, I do like the idea of the analysis as well as the technical implementation. Nevertheless, the choice of words in the manuscript still suggests causal mechanisms that, as I see it, cannot be inferred with certainty from the correlations. For instance, the caption of Fig. 4 says that if B has a minimal delay from A, then B is the first to receive fluctuations from species A. I don't think this is necessarily true, because it could be that A does not "talk" to B at all; both could receive fluctuations from some variable Z at a different delay. To solve this objection, it is probably sufficient to "soften" a few sentences in several places; for instance, "which suggests that species B is the first to receive fluctuations from species A" rather than "so species B are the first to receive fluctuations from species A".

Minor comments (5 -- 9): the authors fixed these issues.

Comments by Reviewer #2:

The authors have done solid work in this this review. My central concerns are now addressed, and I support publication.

We thank both reviewers for their positive response on our revised manuscript.

We followed the final suggestion of Reviewer #1 (comment 4A/B) and softened the language in several places (lines 412-3, 437-8, 456 and Fig. 4 caption). We marked these changes in the attached PDF.